# Physical Activity and Sedentary Behaviour with Retirement in Maltese Civil Servants: A Dialectical Mixed-Method Study

**DOI:** 10.3390/ijerph192114598

**Published:** 2022-11-07

**Authors:** Karl Spiteri, John Xerri de Caro, Kate Grafton, Bob Laventure, David R. Broom

**Affiliations:** 1Faculty Research Centre for Sport, Exercise and Life Sciences, Coventry University, Coventry CV1 5FB, UK; 2Physiotherapy Department, St. Vincent De Paul Long-Term Care Facility, LQA 3301 Luqa, Malta; 3Physiotherapy Department, Faculty of Health Sciences, University of Malta, MSD 2080 Msida, Malta; 4School of Health & Social Care, University of Lincoln, Lincoln LN6 7TS, UK; 5Later Life Training, Northumberland NE65 0BB, UK

**Keywords:** physical activity, sedentary behaviour, retirement, older adult, mixed methods

## Abstract

(1) Background: Retirement is a life event that can influence physical activity (PA) and sedentary behaviour (SB) and can be used as an opportunity to promote positive lifestyle choices. The aims of this study were to (a) to identify changes in PA and SB resulting from retirement and (b) to explore predictors of any changes in PA and SB following retirement in Maltese civil servants. (2) Methods: a hybrid mixed-method (MM) study, using first quantitative followed by qualitative methods, of civil servants aged ≥60 years, who were followed during their retirement transition for two years. A proportion of the research participants in the MM study retired while the others remained employed. Questionnaires and semi-structured interviews were used to collect data. (3) Results: there were no changes in total PA and sitting behaviour with retirement in Maltese civil servants. People who retired carried out more domestic PA compared to when they were in employment, which resulted in more moderate-intensity PA behaviour. People perceived that their sitting time increased with retirement in the qualitative interviews, but this was not observed in the quantitative data. Past PA behaviour was an important predictor of future PA behaviour, but not for SB. (4) Conclusions: A change in PA occurs with the retirement transition. However, the uptake of exercise is a personal choice that is dependent on previous experience. Increasing SB is perceived as part of the retirement plan but is not necessarily seen in the measured quantitative data.

## 1. Introduction

Physical activity (PA) and sedentary behaviour (SB) influences the health of older adults. Engaging in PA and achieving the World Health Organization’s (WHO) PA guidelines aids in the prevention of disease, whilst excessive SB results in an increased risk of noncommunicable disease development and functional decline [1]. Life events can disturb a person’s daily routine; they are considered specific transitions that a person is expected to experience [2]. Throughout life, a person is expected to adapt to new situations, circumstances, and events. Retirement is a life event that is a possible opportunity to promote healthy behaviours [3].

### 1.1. Retirement

Retirement is a social construct that serves social institutions, social groups, and age succession. Being a social construct, retirement changes throughout the years, with bridge employment becoming a common occurrence [4]. As a consequence of these changes, the retirement age of 62, 63 or 65 years is becoming less relevant to studying retirement behaviour [5]. Wang and Shi [6] defined retirement as an individual’s exit from the workforce, which accompanies decreased psychological commitment to and behavioural withdrawal from work. Based on this definition, a self-assessment approach to retirement was taken in this study [7]. As retirement involves a withdrawal from work and a psychological commitment, the person can self-evaluate whether one feels retired or not.

### 1.2. Predictors of Change in Physical Activity and Sedentary Behaviour with Retirement

Retirement is a period of adaptation, that various authors investigated if going through this particular life event has an influence on PA and SB. Three systematic reviews have examined retirement and PA [8,9,10]; a further two examined multiple life events (which included retirement) and PA [11,12]. There is an agreement from this evidence that retirement results in an increase in PA in the leisure domains, and an increase in light and moderate-intensity PA. However, the initial increase in PA will decrease in the long term [12]. Differences on how retirement influences PA varies by socioeconomic status (SES) [10] and gender [12].

Two systematic reviews have specifically examined SB and retirement [10,13]. Based on these reviews, retirement was found to contribute towards an increase in screen time. Vansweevelt et al. [10] found that changes are more favourable for people with higher SES, whom have a positive increase in PA and a lesser increase in screen time. When analysing data from Stockholm Public Health Cohort, Ter Hoeve et al. [14] found that people are likely to continue with similar behaviours of PA and SB when they go through retirement. Few studies have examined people’s perception of how going through retirement can influence PA and SB. From these systematic reviews and further literature search by the authors, a total of four studies have been identified which explore this. Three studies [15,16,17] have examined retirees’ perceptions of how retirement influences PA behaviour. McDonald et al. [15] identified four themes associated with the perceptions of PA with retirement: (1) resources, (2) daily structure, (3) opportunities, and (4) transitional phase in retirement. Van Dyck et al.’s [17] qualitative findings about PA confirmed previous studies, as they found that participants experienced a change in PA behaviour during their early retirement. However, this change varied depending on their previous PA undertaken during their employment. Socci et al. [16] identified six patterns of how the retirement transition might influence PA based on participants’ perceptions. The identified patterns of PA during the retirement transition were: increasers, continuators, starters, decreasers, fluctuaters, and inactive. They described the phenotype of people undergoing these types of patterns.

Two studies were identified that looked at the participants’ perceptions of retirement on SB [17,18]. Van Dyck et al. [17] found that participants felt that retirement influenced their SB. They experienced a decrease in sitting time during transport and engaged in active transportation. Eklund et al. [18] identified two themes on how retirement influences SB: (1) the theme ‘in light of meaningful SB’, related to the meaning of sedentary activities, in which participants viewed SB as a possibility to perform activities that they were unable to when still in employment; (2) the theme ‘in the shadow of involuntary SB’, which was about the unhealthy aspect of SB and factors that contributed towards the development of such behaviour.

### 1.3. Determinants of Physical Activity and Sedentary Behaviour

The study of behaviours is complex due to bidirectional relationships between environmental, psychosocial, and other variables [19]. The Determinants of Diet and Physical Activity (DEDIPAC) Knowledge Hub carried out a series of systematic reviews that examined the different determinants of PA and SB in older adults. Age and education were correlated with SB in people over 65 years of age [20]. Cultural variations were identified, with education being inversely correlated in European studies but not in Asia. These reviews concluded that there is limited research on SB and old age, with studies being of a cross-sectional design. Brug et al. [21] summarised the findings from seven systematic reviews carried out by the DEDIPAC group on the determinants of PA. PA in people aged 65 years and older was correlated positively with self-efficacy, intrinsic motivation, walkability of one’s neighborhood, health status, fitness level, and higher SES; PA was negatively correlated with falls, smoking, and age. Varied correlations were found for temperature and life events. It was highlighted that there is a need for longitudinal studies on the determinants of PA. There was an interaction between different determinants, which made it difficult to identify an effect due to specific factors. Certain behavioural determinants were not consistently measured, which created issues with compiling data from different studies.

An identified gap in the literature pertains to the apparent absence of studies that seek to understand the retirement transition and its influence on SB and PA longitudinally. The other gap in the literature is the lack of integration of the understanding of changes in PA and SB that take place and how people experience these changes. In view of this, the research question for this study was therefore ‘Does retirement influence PA and SB in Maltese civil servants and what are the predictors of change?’ The aims were (a) to identify any changes in PA and SB resulting from retirement in Maltese civil servants, and (b) to explore predictors of any changes in PA and SB following retirement in Maltese civil servants. The findings of this study are a progression from a previously published study focusing on pre-retirement perceptions of PA and SB in the same population [22]. The main findings from this were that the retirement transition is a subjective transition, with people bringing on their experience in this transition and portraying their expectations of PA and SB based on these. The study identified six themes: (1) learned experiences, (2) psychosocial factors shaping the retirement transition, (3) the discernment aspect of retirement, (4) engagement in PA, (5) the inevitable process of ageing, and (6) cognisant SB.

## 2. Materials and Methods

To achieve the aims, a two-year follow-up study using mixed methods (MM) based on a dialectical philosophy was undertaken. The dialectic philosophy is a form of dynamic logic of thesis, antithesis, and synthesis [23]; it provides a framework as to how two epistemologies are used together for a deeper understanding of the phenomena under study. A hybrid MM design was adopted (QUAN→QUAL→QUAN + QUAL); quantitative data were collected first, followed by qualitative data from the same sample. The quantitative and qualitative data were recollected after two years. QUAN data were collected as part of a cohort study, based on a postpositivist methodology, which recruited Maltese civil servants who were ≥60 years old to be monitored using surveys. In the QUAN data collection, the researcher interaction with participants was kept to a minimum to reduce biases. The QUAL data were collected during narrative interviews, based on a socioconstructionist methodology, with participants who were planning to retire within the subsequent 6 months to 1 year from the survey. Using a socioconstructionist methodology in the data collected, it was assumed that multiple realities exist and knowledge is context-specific and value-laden. Inclusion criteria for the cohort study were Maltese civil servants aged 60 years or over, who were able to speak in Maltese or English, and for the interview, participants had to be planning to retire and willing to participate. Figure 1 represents the research design and where integration of the strands took place.

### 2.1. Data Collection

Data for the QUAN strand was collected using a questionnaire which included the Maltese version of International Physical Activity Questionnaire—long version (IPAQ) [24,25], the Exercise Motivation Inventory—2 (EMI-2) [26,27] and open-ended questions about barriers towards PA in the past 2 weeks [28]. The translated Maltese versions of the IPAQ and EMI-2 were used as they were found to have good reliability, like the original English version [24,26]. Demographic information was collected on marital status, education, sex, and employment scale. In the first data collection, participants were asked if they were planning to retire within the coming year, whilst at follow-up they were asked if they were retired or not. The definition of retirement was based on self-assessment, where it was based on whether participants felt retired or not [7]. The questionnaires were available in English and Maltese. The QUAL data were collected via a semi-structured interview using open-ended questions. These were guided by the theoretical domain framework (TDF) [29]. The TDF was developed with the aim of improving the use of behaviour change theories by creating a framework that integrates concepts from various theories [30]. The framework consists of 14 domains, each of which has various constructs; examples of domains are knowledge, skills, and environment [31]. The interview guide was piloted both in English and in Maltese. Participants were asked to tell their life stories during the interview. The second interview guide is available in Appendix A. Figure 2 is a schematic to highlight participant recruitment and completion for the study.

### 2.2. Recruitment

Participants were recruited via an email sent to all Maltese civil servants who were 60 years of age and over. The email included a recruitment letter with links to the study survey. Interested participants completed the consent form and the questionnaires either online or provided details to be undertaken via telephone. Those participants who were planning to retire within the subsequent 6 months to 1 year were asked if they were willing to participate in an interview about their retirement experience. Purposeful sampling using IPAQ categories to select participants with different PA behaviour was used to recruit participants for the interview with a minimum quota set at 20 due to the two-year follow-up period. The initial data collection was carried out between September 2019 and January 2020 and the follow-up was undertaken between October 2021 and January 2022. Months were kept similar to avoid seasonal variation.

### 2.3. Ethics

Participants were given an information letter about the study and provided written consent to participate each time a request was made to complete the questionnaires and/or undertake the interview. Participants had the right to withdraw from the study and were provided with an opportunity to ask questions pertaining to the study at any point. Ethical approval was obtained from Sheffield Hallam University Research Ethics committee reference number: ER9249191 and Coventry University ethics reference P115641 due to a change in host institution.

### 2.4. Data Analysis

Data analysis was carried out separately for the two strands but interpreted concurrently with equal weighting to achieve data integration as one of the MM criteria. For QUAN, the questionnaires were analysed based on the guidelines provided for IPAQ and EMI-2. Barriers were analysed on whether participants identified barriers towards PA or not and the type of barriers identified. Descriptive statistics based on mean and median were carried out. Inferential statistics, which checked for differences between sex, education and retirement status, were undertaken using an independent *t*-test and chi-square or their nonparametric equivalent following checking for normal distribution. The analysis was carried out separately for the initial and follow-up stage of the data collection and then differences where checked between the two time points using Wilcoxon. Statistical significance was accepted if *p* < 0.05.

Data analysis for the QUAL data was carried out using reflexive thematic analysis [32] and structural narrative analysis [33,34,35]. Reflexive thematic analysis was based on steps by Braun and Clarke [32]: (1) data familiarisation, (2) generating initial codes, (3) searching for themes, (4) reviewing themes, (5) defining and naming themes (6) report writing. The structural narrative analysis included five steps [33]: (1) the first step was to identify interviews that had good stories; (2) this was followed by developing the story structure, which was based on the work of Labov and Waletzky’s (1972); (3) then it was identified why the story was being told; (4) stories with similar structure, content, and meaning were grouped; and (5) report was written.

The Initial analysis of using bot” app’oaches was carried out by KS using the manuscript in the original language. This was then discussed with JXDC over multiple meetings. Once meaning was agreed, this was discussed with the rest of the authors. Transcription was carried out by KS in the original language of the interviews; the five transcripts were checked by an independent researcher to check for consistency with transcripts.

Data were then integrated using joint display [36,37]. In using joint display, the data from the two strands were displayed together to check for consistency and identify any differences in the data and analysis carried out. Convergent (similar data from the different strands) and divergent (conflicting evidence from the different strands) findings were identified using a traffic light system: amber was used for areas where further research questions were elicited, green where there was agreement (convergent) between the data, and red where there was disagreement (divergent).

Rigour for this study considered the QUAN, QUAL, and MM aspects. For the QUAN strand, the tools used to collect data on PA [25,38] and exercise motivations [26,27] were reliable and valid. The study attempted to recruit participants from all the population to attempt generalisability for the population being examined. In the QUAL strand, the lead researcher kept a reflective diary throughout the process, which aided in the data interpretation and for reflexivity. Raw data are presented as supplementary material for transparency (Appendix B). Participants were engaged with a high response rate at follow-up. Their experience was shared as they went through the retirement process and a rich description was provided for all participants. Data analysis triangulation was used for an in-depth understanding of PA and SB. To ensure quality for the MM aspect of the study, the study was aligned to a dialectical philosophical approach. Integration of MM took place at the point of sampling, results, and interpretation (Figure 1). The results from the QUAL and QUAN were integrated and illustrated using joint display.

## 3. Results

The response rate for the initial survey was 11% (*n* = 96) of the targeted population (*n* = 872). The IPAQ was incomplete for 7 participants reducing the number of useable questionnaires to 89. From these, 20 participants were purposefully recruited for narrative interviews. At follow-up, the response rate to the survey was 48.3% (*n* = 43), whilst that for the interview was 95% (*n* = 19) (Table 1). There was no statistical difference at the initial survey between respondents and nonrespondents in: PA, sitting time, sex, education, marital status, and barriers towards PA. There was a statistical difference in 3 motivational constructs out of 14 from the EMI-2: revitalisation, challenge, and affiliation with a higher mean in those who completed both surveys.

At follow-up, there was a statistically significant difference in sitting time between retired and nonretired participants (*p* = 0.010), with nonretired participants having higher sitting times (Table 2). Other PA domains were not statistically different. When checking for differences between the first and second data collection, those who retired had a statistically significant change in moderate-intensity PA (*p* = 0.037), whilst the nonretired had a change in walking PA in MET min per week (*p* = 0.023). Those who retired experienced a statistically significant difference in the PA domains of work, domestic, and leisure; the nonretired only experienced a statistically significant change in leisure PA (Table 3). There were no statistical changes to motivation in the retired, whilst the nonretired experienced a decrease in motivation in the challenge domain.

### 3.1. Reflexive Thematic Anlayis

Follow-up interviews lasted, on average, 39 min in duration and ranged between 16 min and 1 h and 25 min. A total of 12 participants had retired and the other 7 continued to work within the same employment. After the initial coding phase of the reflexive thematic analysis, 46 codes were identified, which were finally represented within two main themes. The themes and subthemes with supporting quotes are presented in Table 4 and Table 5.

#### 3.1.1. Theme 1: The Impact of Official Retirement Age Is Inexorable (Inevitable)

Reaching the mandatory retirement age brought about changes to the individual that were not influenced by whether a person retired or continued to work. In the follow-up interview, participants who did not retire were able to articulate their retirement plans in more detail compared to two years prior, even though some still had not chosen the day to officially retire. When retirement age was reached, the person was able to decide whether to retire or not, depending on their wishes and personal commitments at that time. Reaching retirement age influenced individuals irrespective of their decision to retire.

#### 3.1.2. Theme 2: The Retirement Plan Is Influenced by Circumstance during the Transition

Retirement plans develop during the retirement transition, as persons focus on their intentions once they reach retirement. When nearing retirement age, it was found that some people had fixed plans whilst others did not. Once the decision to retire was consciously taken, any plans for physical activity and sedentary activities were reflected upon and understood to be influenced by a number of factors, including: (a) specific circumstances, e.g., if one’s partner was working or not, if children were living at home or not, finances, and self-health; (b) experiences, e.g., culture of exercise participation, physical-activity involvement, and impact of retirement experiences by significant others; and (c) intentions, e.g., gardening/farming activities, meeting with friends, and starting exercising. The COVID-19 pandemic was an important aspect and was regarded as an unexpected interloper in these persons’ retirement plans, for which sudden and unexpected adjustments had to be made. In dealing with the retirement transition, participants used their varied resources to adapt to the new situation. Some participants highlighted the difficulty in adjusting to this. This theme and its subthemes, with supporting quotes, are presented in Table 4. Original quotes are provided in Appendix C, Table A2 and Table A3.

The changes between the different data collection points were analysed by merging the two reflexive thematic analyses from the previous paper [22] and those presented in the current one. The results are present in Table 6; the themes at each data collection point compared and contrasted and changes identified are discussed.

### 3.2. Structural Narrative Analysis

The structural narrative analysis identified patterns in how participants experienced their retirement transition with regards to the PA and SB. Four interviews, whereby participants provided narrations of their experience, were chosen. These specific four were chosen because the interviewees shared multiple stories to explain their experiences. Analysis was carried out on these interviews’ extracts with coding to develop the story structure [33]. Two narratives from participants who had retired around the same period are presented in Table 7, which allows for comparison. Names have been changed for anonymity purposes. When examining their retirement stories, their narratives were similarly structured but had different PA and SB outcomes. Results are presented in Table 7; supporting quotes and participants’ reconstructed stories are available as Appendix B.

### 3.3. Data Integration

The aim of using a dialectical MM approach was to integrate the data from the two strands and achieve an understanding of how the retirement transition can influence PA and SB. Data integration was carried out using a joint display (Appendix D) and flowchart (Figure 3). This study identified six meta-inferences when integrating the different data. Three meta-inferences agreed, one was in dissonance, and two showed diffractions (Table 8).

## 4. Discussion

This study adds to the body of knowledge about the perception of people going through the retirement transition and how it can influence PA and SB. The use of MM highlights the differences in what people perceived and what they reported to have happened to their PA and SB. The study highlights the difficulty that retirees experience in adjusting to a new routine and the need for them to be supported in this transition.

### 4.1. Decision to Retire

The decision to retire was present in the participants’ life story. It was inferred to—but not presented as—a particular day in their story line. It was a process that could have started with the discernment aspect in the preretirement period. This fits with the retirement theories that conceptualise retirement as a decision-making process [6,44]. As hypothesised by this theory, participants consider opportunities and take an informed decision on whether to retire or not. This finding also supports the role theory; as the participants started reaching retirement age, they started to take decisions to change their role from an employee to retired [45]. The timing of deciding when this change in role happens can be influenced by various psychosocial factors such as family, finance, and health [46], which were not assessed QUAN in the study. Going through retirement does not necessarily mean that a person wanted to retire. There were two participants who had retired but their decision to retire was not theirs, and they wanted to return to work. Those who took a conscious decision to retire did not want to go back. Bridge employment is an option that some employees consider as their exit strategy from the workforce [47].

### 4.2. Replacing Work Physical Activity

The replacement of former work-related PA and SB with other means of activity was apparent in both data sets. The inevitable change is identified within the statistical change in Domestic and Work PA in the retired compared to nonretired participants. Using the QUAN results, the retired group showed an increase in their PA domains across leisure and domestic PA, whilst those who did not retire had only a change in their leisure time PA. However, using accelerometer measurements, it was found that work PA was replaced by sedentary activities and not necessarily by other PA, especially in people with a manual occupation [39]. These changes can be determined by various factors such as psychosocial factors, financial situation, and health status [21], which were not assessed in this study. Socci et al. [16] had found that retirement can have different influences on PA depending on identified phenotypes. In the QUAL data, this change is highlighted by retired participants claiming to try and fill the empty time with other activities, especially in the morning. The QUAL highlights how this change was challenging for the participants and was one of the least considered. These changes are documented within the literature with studies finding changes in domestic and leisure time PA over the retirement transition [8,12]. Retirees might be more inclined and worried about the financial aspect of retirement rather than the social and activity part [47]. This might explain the difficulties that retirees experience in adapting to the change in activity patterns, and provides further insight as to why studies [12] might have found different conclusions as to whether leisure time PA increases or not. There is a difference between retirement planning and effecting the plan when retiring [48], as highlighted in the participants’ stories their activity patterns were influenced by their previous experiences and perceptions. The continuation in PA patterns was found in other studies [14,16].

### 4.3. Changes in Physical Activity Domains

As part of the adjustment process, the retired group demonstrated changes in the different PA domains, with the exception for transport. Other studies have found that there are changes in active transportation; however, in this study, the PA in the transport domain pre-retirement was low, which might explain why a change did not occur. There was a statistically significant change in moderate types of activity within the retired individuals and a statistically significant change in walking PA in the nonretired. A statistically significant change in leisure time PA was found in both groups. The change in PA intensities within the retired participants could be part of the adjustment process and them developing a new norm. When retiring, people try and find meaningful activities [49], which is found in what they valued previous to their retirement. Participants try to maintain the same activities but now they experience more time. As PA was measured using self-reported data and PA intensity is perceived, not actual [50], this could explain the change in moderate PA identified. In addition, most of the participants had low PA jobs prior to retiring, even those who had low education levels; this could have led to an increase in moderate PA by engaging in domestic PA. This uptake in PA related to domestic work was present in most of the interviewed retired participants.

### 4.4. Changes in Leisure Time Physical Activity

Changes in leisure time PA were present in retirees and those still in employment. In addition, there was no statistical difference between the groups. The whole population had a change in their leisure time PA. These results fit within the participants’ story structures, whereby they projected their current state of PA into their retirement plan. The changes at the population level in leisure time could be due to COVID, ageing, or other psychosocial factors. Retirement does not change what people value in life [51]; there is no reason to believe that retirement will change people’s perspectives on exercise, physical activity, or sedentary behaviour. Lifestyle choices taken pre-retirement are likely to persist post-retirement [14]. The retirement transition provides an opportunity where people have more time to engage in exercise; however, if this is not part of their culture, it might not be what they would have envisaged for their retirement. As found by Vansweevelt et al. [10] in their systematic review, with retirement, people in higher SES tend to undertake more favourable changes to PA domains and sitting time.

### 4.5. Sitting Time

Statistically a significant difference in total sitting time was present between the retired and nonretired, with the latter having a higher sitting time. Using device-based measures, it was found that women increased and prolonged their sitting time with retirement, but not men [42,43]. However, men start increasing their sitting time prior to retirement [43]. This study was unable to make this distinction due to the sample size. Participants in this study claimed to be developing a new norm with different patterns. The different patterns being created might be related to the type of activity being carried out, but not to the amount of activity being carried out. Exercise behaviours might have purpose for retired older adults, whilst other types of activities such as meeting friends might have purpose and happiness [52]. New norms might be developed based on what was valued previously. Sitting behaviours might be based on previous behaviours, explaining why a change was not found within the retired participants.

### 4.6. COVID-19 and Other Factors

Participants claimed that the COVID-19 pandemic had an impact on their lifestyle. Some of the changes identified seem independent of the retirement transition. The pandemic could have had an impact on the participants’ behaviour in different ways based on psychosocial factors and impacted the retirement plan. Some of the identified changes were present within the population, but not when analysing by retirement status. This shows that there were other impacts that need to be considered during the retirement transition, not limited to the shift from work to retirement. Another factor that could have influenced the lifestyle changes and was not assessed in this study was SES, which was found to influence PA and SB during the retirement transition [10]. Factors such as self-efficacy and environmental factors [19] were also not included in the data collection.

### 4.7. Strengths and Limitations

The use of a two-year observational approach enabled the in-depth study of PA and SB over time, and thus captured the influence of retirement on these variables in a manner that to date has been missing from the literature. This should be considered a key strength of the present study and represents a novel and original contribution to knowledge. Another strength of the study is the integration of a two-year observational and narrative study. The participants were followed during their retirement transition, which provided an opportunity to integrate experiences and opinions with changes to PA and SB. The integration of these health behaviours with the retirement transition provided a deeper understanding of how such behaviours can be targeted in future interventional studies.

One of the limitations was the low retention of participants in the observation study; however, there was no statistical difference at the first data collection between drop out and those who participated at both data collection points. The small sample size did not allow for adjustment for gender and SES, and for generalisability of findings to the whole population. The results of inferential statistics should therefore be interpreted with caution.

The use of self-reported measures was another limitation. Participants were provided with an opportunity to use reliably tested tools in two languages to improve participation and include people from low socioeconomic classes. The number of participants with secondary education was 18%, and only 1% had primary level education. This might be due to recruitment method used, as this was conducted through the official government email address, which might not be used as much in people with low education, but this is highly speculative. The participation in the narrative study was high, which shows participants’ engagement with the study. Using data analysis triangulation provided an in-depth understanding of the participants’ experiences.

### 4.8. Practical Application

The study highlights the need to consider PA and SB as part of the retirement adjustment process. It identifies the importance of considering previous experiences when promoting health behaviours during this period to support a healthy adjustment to retirement. There is a need to support people going through retirement to adjust to their new routine and prepare them for these changes. Perception about ageing needs to be considered when people are going through the retirement transition, as these can influence retirement adjustment. Retirement planning interventions to promote health behaviours and assist employees in adjusting to their retirement should be considered. Employers can use the retirement transition as a positive opportunity to support retirees.

## 5. Conclusions

The main finding from the study is that undertaking exercise and being active is a choice within the retirement transition. PA is a consequence and increasing SB is integrated within the retirement plan. Retirement brings about a change in PA and SB as the working hours need to be substituted with other activities. However, the impact of the retirement transition on PA and SB might be limited as people revert to their known habits and experiences to fill the void left by retirement. Retirement presented an opportunity for those who wanted to exercise further consciously. Concurrently, people are required to change their PA patterns due to the implications of stopping from work. SB is part of the retirement plan, as people expect sedentary behaviours to be part of the retirement activities. Interventions that target behaviour during the retirement transition must address the retirement process, starting at the preretirement phase when people are discerning about their retirement. Health behaviours are often considered of secondary importance during the retirement transition by individuals, and yet changes in their PA engagement and patterns will inevitably happen because of retirement. A critical suggestion arising from this research—and one which has not been forthcoming from prior work on this topic—is that any intervention needs to consider the retirement transition and not focus solely on health behaviours.

## Figures and Tables

**Figure 1 ijerph-19-14598-f001:**
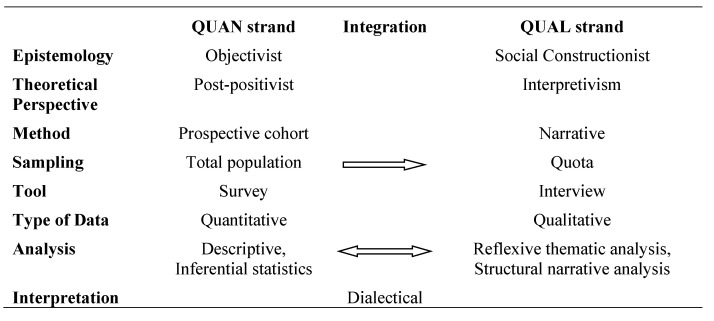
Research design of dialectical MM.

**Figure 2 ijerph-19-14598-f002:**
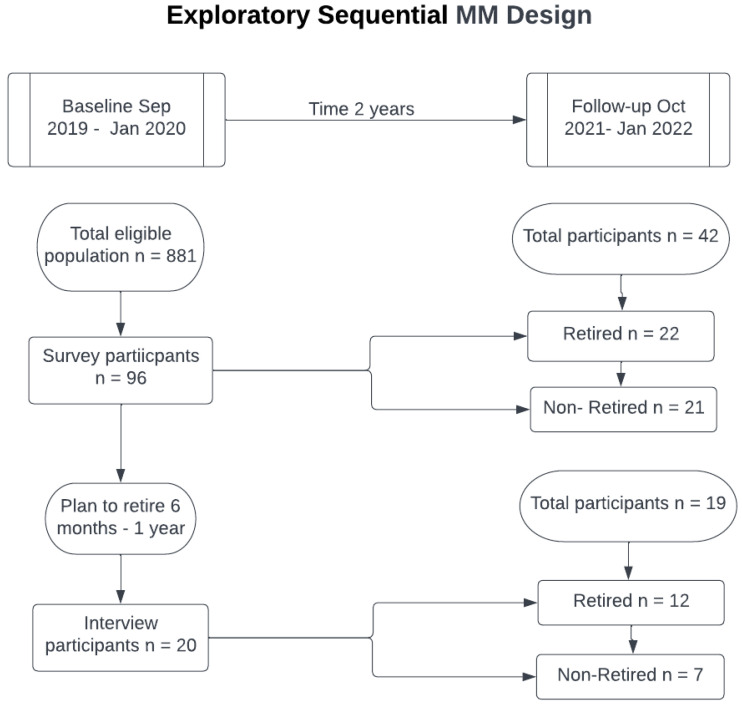
Diagrammatic representation of the adopted MM design.

**Figure 3 ijerph-19-14598-f003:**
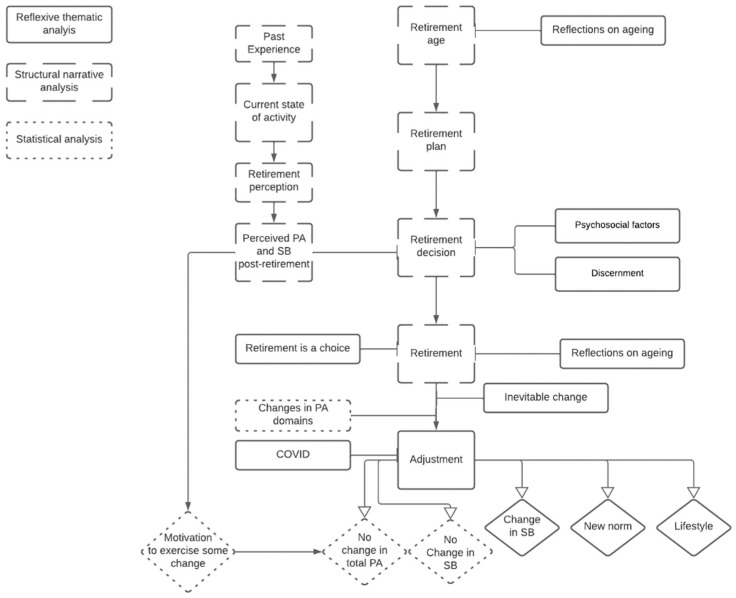
Data integration flowchart.

**Table 1 ijerph-19-14598-t001:** Demographic description of the population.

	Variable	Initial Population		Population at Follow-Up	
		*n*	%	*n*	%
Sex	Male	44	49.4	20	46.5
	Female	45	50.6	23	53.4
Status	Single	7	7.9	3	6.9
	Married	76	85.4	35	81.4
	Widow	1	1.1	1	2.3
	Lives with partner	5	5.6	4	9.3
Scale	(1–6) Top management	44	49.4	18	41.9
	(7–10) Middle-management	31	34.8	14	32.6
	(11–15) Other	14	15.7	10	23.3
Education	Primary	1	1.1	1	2.3
	Secondary	16	18.0	11	25.6
	Post-secondary	24	27.0	16	37.2
	Tertiary	48	53.9	15	34.8
IPAQ PA categories	Low	21	23.6	13	30.2
	Medium	34	38.2	17	39.6
	High	34	38.2	13	30.2

**Table 2 ijerph-19-14598-t002:** Differences in PA and SB by retirement status.

Retirement Status		Retired (*n* = 22)			Nonretired (*n =* 21)		Mann–Whitney Test (Comparison between Groups)
	25th	50th	75th	25th	50th	75th	(*p*-Value)
Total sitting time per week (min)	1050	1680	2340	1680	2520	3360	0.046 *
Total PA (MET min per week)	1406	2995	5628	1512	2735	4523	0.521
Total vigorous (MET min per week)	0	0	0	0	0	240	0.349
Total moderate (MET min per week)	480	960	5040	60	630	1800	0.040 *
Total walking (MET min per week)	198	495	2030	297	1089	2376	0.502
Total leisure (MET min per week)	198	891	1390	0	554	1040	0.365
Total domestic (MET min per week)	440	960	4800	60	540	1080	0.071
Total transport (MET min per week)	0	0	396	0	231	792	0.264
Total work (MET min per week)	0	0	0	0	269	990	0.000 *

* shows statistical significance.

**Table 3 ijerph-19-14598-t003:** Mean change in sitting and PA domains within group.

	Retired (*n* = 22)		Nonretired (*n* = 21)	
Variable	Mean Change	(Wilcoxon)	Mean Change	(Wilcoxon)
Sitting (min)	−289.09	0.444	127.50	0.406
Total PA (MET min per week)	−1481.63	0.117	−823.26	0.084
Vigorous PA (MET min per week)	−221.82	0.799	−434.80	1.00
Moderate PA (MET min per week)	625.45	0.037 *	−469.13	0.408
Walking PA (MET min per week)	226.60	0.546	454.58	0.023 *
Work PA (MET min per week)	−396.09	0.003 *	−275.00	0.778
Domestic PA (MET min per week)	770.91	0.028 *	−572.12	0.836
Transport PA (MET min per week)	15.94	0.600	187.35	0.897
Leisure PA (MET min per week)	−396.09	0.009 *	210.43	0.019 *

* shows statistical significance.

**Table 4 ijerph-19-14598-t004:** Subthemes of theme one, ‘the impact of official retirement age is seemingly inexorable’, with supporting quotes.

Subtheme	Quote in English
(1) Change is inevitable The retirement process presents the person with a reality by which present/past experiences will differ from present/future experiences. This is the change that takes place as a consequence of retirement. All this is inevitable, as it is part of a conscious decision leading to retirement that also brings with it a level of change in physical activity and sedentary behaviour, a consequence of which life adjustments are made.	“*…during the day you try *[your best]* …understood. For example, my wife still works, and the children still live with us, so I try to help with the housework. Then, you must take on certain responsibilities. For example, do some housework, errands and things like that. Some hobbies. Maybe I’ll stay in the garage and do some work there, things like that. Are you understanding me? It fills the day like this. Day by day*”. —Chris
	“*…It’s a new beginning. I enjoyed what I had, but now it’s time for a change. And it is not easy ‘cause you have to think *[about what to do]*. But I was preparing myself. I think, when I spoke with you, I already had a plan of things. And, in fact, I didn’t do *[as she was busy with other things]*. A lot of it was fear that I was not going to be able to adjust. I’m going to have to, too much time on my hands and I’m gonna get bored. And if Lilly is bored, uh, and uh *[that’s a problem]*. So yes, uhm, it’s still there…*”—Lilly
(2) Retirement is a choice As the person reaches retirement age, a shift in a locus of power was noted. It ostensibly results in the control of one’s decisions, the decision to retire being the individuals. It is the legitimisation of retirement that offers the person the power of control based on an implicit right (legitimising the process), therefore making the decision to cease employment (and hence retire) or to continue working a personal choice. There are different factors that will influence this decision and on which the influence of this decision would have an impact, including physical activity and sedentary behaviour. Indeed, those who decided to continue working reported that the legitimisation of their decision to continue working, coupled with their experience, gave them an added advantage to manage their lifestyle and handle situations in a different way.	“*…I had an opportunity to receive the pension and stay on. I said I’ll take it *[the opportunity]* because it won’t come again, I don’t have any difficulties as a security guard, I said. And I said, yes, I’ll do it, so it’s been two years since, now. But up to a point I was determined to start the PRL *[pre-retirement leave]*, as some do not take it to get the allowance because we have *[we are entitled to]* an allowance*”—Jason
	“*…I asked for a transfer and was accepted, but then I asked myself where I was going* [with life]. *I said I’m leaving now* [retiring]. *It’s my time*”—Chris
	“*emm before I took the decision to retire, because I took the decision to retired, even though James* [the husband] *has been telling me to leave, leave, leave*”—Agnes
(3) Retirement brings about reflections on ageing Retirement allowed time for reflection, including on ageing, as persons now had the opportunity to reflect as they found themselves having ample alone time. They reflected on their physical and mental abilities, and how these seemingly declined in comparison to when they were younger. As a result of this reflection, retirees took different decisions, and not everyone opted to engage in healthy behaviours.	“*…I can’t say that I still have the health of a 30-year-old, at 64, I have my limitations but I don’t let them keep me from doing things that make me feel happy. There are people who want to go for a run for example, I love to go for a walk, a run takes my breath away, hehe*”—Carmen
	“*…because I’m always thinking. I think about the things I used to do before. You get used to a routine in your life. You have periods in life, like having seasons.* [giving an analogy] *I am currently doing this. It’s time for potatoes for example, so I focus on potatoes. If it’s snowing, you will be thinking about snow. I have permission to sail my boat from May to September but it is not allowed to go on a boat during this period. It’s like you have a stages* [in life]. *Until now I’m still living these experiences, but always with difficulty*”—Jason
	“*I’m feeling tired I don’t have that energy like before, I tell myself I can’t even imagine now how I did all that work, going up and down* [walking from one place to another during work] *and working, because I used to do a lot*”—Jessie

**Table 5 ijerph-19-14598-t005:** Subthemes of theme two, ‘the retirement plan is influenced by the transition’, with supporting quotes.

Subtheme	Translated Quote
(1) Retirement leads to adjustments of physical activity As a consequence of unavoidable changes consistent with retirement, people took to new situations by adapting, including adjustments to their physical activity patterns. These were found to be influenced by the persons’ considerations on (a) new opportunities, (b) the need to remain ‘active’ or to exercise, and (c) age vs. exercise.	
(a) New opportunities—retirement resulted in different perceptions towards new opportunities to be physically active that were in turn influenced by preconceptions on what physical activity really was. The adjustment to being more physically active was at times unexpected, e.g., spending more time on domestic activities, whilst other adjustments were part of the person’s plan, e.g., starting to go for a walk regularly. Going into retirement brought about opportunities for a person to be physically active in different physical activity domains (domestic, transport, and leisure).	“[talking about gardening] *This year, I struggled and suffered, and I only have trees. I have eleven trees in the inner part* [of the garden, and somewhat sheltered] *and another ten on the outer part* [exposed to the elements], *as well as five olive trees. I have never watered the five olive trees because it is impossible* [too much work]. *How much water can I carry? I’ll water four trees one day* [and water others another day], *because you cannot do everything at once. With this heat, it becomes too much. Even if you go in the early evening, it is still too hot. The heat is tremendous*”—Albert
	“*Y*es, *even for shopping, I don’t take the car and go shopping. I prefer going on foot when I only have to shop for a few items. The grocer’s is four blocks away, so it is a good walk. I don’t like to use the car when I can go on foot*”—Jessie
(b) The need to be active or to exercise—Engaging in activities with a purpose was something that all participants sought, whether in retirement or having continued working after their retirement age. Participants reported the need to engage in activities that kept them mentally and physically busy. Retirement brought with it a certain lack of purpose, which participants felt the need to address with meaningful activities. Because they found themselves with more free time, and as they wanted to fill their day, some used this as a motivation to increase their exercise participation or engage more in domestic activities, whilst others engaged in sedentary activities.	“*At the moment, I am working on a few projects. For example, I have some paint and I am painting weathervanes now, green, and white. If you want to order one *[let me know]*. So far, I’ve done two, the first to pass the time but now I am taking it more seriously, so this one is harder work. There are no *[written]* instructions on how to do these, so I’m watching YouTube videos instead. The videos are not instructional, but you just see completed weathervanes in operation. Then, I try to figure out how they work. I am designing one with a man and woman on a seesaw*”—Albert
	“*I go for a walk for half an hour or 45 min. If I can’t do half an hour straight, I go 20 min at one point of the day and 20 min later. But I want to walk, even if I cannot go out that day and stay home, I’ll go up on the roof to hang up the laundry and, then, I’ll walk around the roof. I want those 20 min of walking, free from chores. Then I’ll go back to continue. Hehe*”—Carmen
(c) Age and exercise—retirement, as well as increased exercise behaviour, brought about reflections on ageing and the ageing body. When engaging in exercise, participants were conscious and sensitive to their ageing bodies. They were aware of the need to exercise, but at the same time, they found that it was taking them longer to recover when engaging in exercise activities. Although this did not stop them from exercising, it made them adjust the manner in which they performed exercise.	“*The drive to exercise has to come from within, and time for exercise you have, but you need the physical ability to exercise. You are not as healthy as before. You don’t feel the same way. You get some aches here and there. They start coming. Either your knees or somewhere else. So your ability to exercise will not remain the same. You can modify i.e., you can modify it *[the exercise you do]* but your strength and energy will not remain the same as they used to be. Maybe you do not notice much change just one year after you retire but, as more time passes, you start to realise that you are no longer as adventurous as you used to be. Where, before, you used to get to there, today you get only to here*”—Sean
	“*I still have life commitments. I still don’t have those freedoms *[which other retirees do]*. I’m still healthy, so I can’t *[refuse to help my children]*, I have no excuse for not doing anything*”—Chris
(2) Retirement brings about lifestyle adjustments Retirement led to lifestyle changes (food, consumption, finance management, time for self, smoking, being lazy) that could influence the health of individuals and therefore their physical activity and sedentary behaviour patterns. There were reported changes to food consumption, specifically the consumption of healthier meals as well as eating more frequently. A change in the management of finances was also reported, with people willing to spend money more freely on health or leisure activities. This was linked to their life expectancy, with them perceiving having less time to enjoy their financial gains. Retirement itself, as well as for those who continued to work post-retirement age, provided the opportunity for time to be dedicated to oneself and to do things which they enjoyed. It was also reported that retirement resulted in an increase in the consumption of cigarettes. A lack of structure to daily activities when compared to the work routine brought with it more hours sitting down performing fewer activities. This was perceived as being lazy.	“*Before, I used to tell him *[my husband]*, to do something and we would do it, by car if not on foot. He used to get up and do it. Now, he procrastinates*”—Josette
	“*So, when I won’t be working anymore, I will have more time to organise my day at my own pace. I will have more time for myself, and more time to go swimming*”—Antoinette
	“*…maybe I’m eating more now. I’ve gained weight, somehow. I am not sure how. I have ended up in the kitchen the whole time, because we eat in the middle of the day and in the evening, and I nibble occasional snacks, too*”—Jessie
(3) Retirement leads to the development of a new norm Adjusting to retirement leading to a new norm was reported to be characterised by three influences: (a) the grievance of missing the work environment, (b) a honeymoon period, and (c) a long retirement.	
(a) A common theme emerging with participants who decided to retire was their grievance about missing their work environment, especially the social interactions at work. The specific impact of the COVID-19 pandemic seemed to play an even greater role in this grievance, as maintaining meaningful connections with work colleagues after retirement was more difficult. Retirees missed their work colleagues and the social aspect of work. This was highlighted as an important aspect of working, yet at the same time, they were conscious that once the decision of retirement was taken, they did not want to go back to working.	“*Oh, I miss colleagues. I miss the children I used to work with. I really do. To me, it was rewarding. Maybe it is not that special, being an LSE *[learning support educator]*, but, you know, everyone’s got a role, and I had a lot of job satisfaction, seeing improvements in children and making connections with children and staff. I miss the staff as well, but not that much, because I am not sitting around all day thinking about the past. I’m doing other things now, so my mind is occupied. I still miss work sometimes, but I don’t look back. I don’t want to go back to that, although I miss it. I miss my work. I miss the children and I miss the colleagues as well, but not that much. I am not sitting around telling myself how much I wish I stayed working. No, no, no, no. That, I don’t say. Certainly not*”—Lilly
	“*…I was proud of my work, so I miss that. Nothing more, not because I used to meet friends. I only miss about two of my colleagues out of two dozen, but mostly I miss the satisfaction my work brought, because it is satisfying to help people. That is what I miss*”—Claire
(b) The initial part of the retirement was defined as a honeymoon period. Participants described this as being a period of long leave or vacation. The duration of this honeymoon period varied amongst the participants, yet it was always described as being a happy period of retirement. During this time, participants either engaged in activities that they had planned to carry out in advance, or decided to rest by taking it easy, relaxing, and being more sedentary.	“*I retired, but there were still things I needed to do and settle, especially in the first few months. There are finances, paperwork and some projects I had always wanted to work on but never had the time for. So I had things to do, but also more free time to do them. If I suddenly decided I wanted to do something, I could just go and do it. There is nothing hindering me. I have been freer to meet my daughter*”—Agnes
	“*The first day I was so happy that I left. I didn’t have anything to do. I did not know where to start, what I was going to do. I watched television. Yes, now I can watch television. The first two days, three, television movies, my son got me Netflix. After a week, I started feeling restless. I increased my food intake. I stopped reading and ate more. After about a fortnight of watching television, I found myself eating a plate of pasta at half past eight in the morning. How good is that! But I cannot go on like this. I’m going to become the size of a mule if I keep going on *[eating]* like this*”—Albert
(c) As the honeymoon period weaned off, retirees started to grasp that this would be their new life. This brought about anxieties as well as a conscious effort to attempt to fill up the day with various activities.	“*I didn’t know what to do at first so I started working around the house a lot. Then, my grandson *[was born]*. It was a new thing having him. But, then, it *[retirement]* hit me. I started to feel empty, bored and sad with all that free time. The day was not satisfying enough. The last few days, I do not even have the appetite to get up. Before, I used to get up in the morning and go for a walk. Now, I don’t have it in me. I don’t have the will. I don’t have the strength to get up anymore*”—Agnes
(4) Sedentary behaviour is influenced as an impact of retirement The morning was identified as the time of day when retirees reported trying to be more active. Sedentary behaviours were more attributed to the afternoon. Activities with a specific purpose, such as performing desk jobs or maintenance activities or using the computer, were carried out while sitting, as well as activities with little purpose, such as watching television when there was nothing better to do. Those who retired from a primarily sedentary job noted that on retirement, they carried out domestic types of activities, and as a consequence, sat less when compared to their previous work routine.	“*I start to see myself strangely, because I eat and read, then I watch some animal documentary on television or something like that, or I watch the BBC, and maybe I snooze for a quarter of an hour. Then I wake up, prepare a flask of coffee and go and play with the dog*”—Albert
	“*My exercise routine remained the same. Maybe it got worse. Instead of finding more time for it, sometimes I get lazier. Now, I have all the time I need for it but I keep putting exercise off. I keep thinking of increasing my exercise, so the desire is there, but my will seems to have left me*”—Sean
	“*…When I feel I’m sitting down too much at the computer or whatever, I’ll put the music on… put the blinds down, and I dance*”—Lilly
(5) An unexpected pandemic influences the retirement transition The retirement transition period examined was between 2019 and 2022, and therefore was influenced by the COVID-19 pandemic, which was unplanned and impactful. The adjustments to the type of physical activities and sedentary behaviours, as noted by the retirees, was not only impacted by retirement, but also by COVID-19. Some retirees pushed their retirement day forward, as COVID-19 adjustments made their employment terms more favourable to stay on; while others terminated their employment, even though they were considering a work extension prior to the outbreak. The pandemic had some influence on the amount of physical activity and sedentary behaviour patterns, which were short-lived. Those inclined towards exercising claimed to have found that the pandemic provided an opportunity to increase their outdoor activities. The restrictions imposed on social interactions also provided an opportunity to retirees to discover the outdoors with less traffic in the streets. Those who were less motivated towards exercise, or who enjoyed exercising within a social environment, identified the COVID-19 restrictions as limiting their exercise participation. The lack of social interaction provided an opportunity for some to engage in new hobbies that they had not considered before.	“*I think, before I stopped, I was thinking that, maybe two days a week, I will visit the day centre *[as a part-time work]*. But, as things turned out *[because of COVID-19]*, I did not succeed. Today, I do not consider that as an option anymore*”—Claire
	“*When I started working from home *[because of COVID-19]*, I started walking more. In the morning, I used to get up early to avoid meeting anyone. I used to walk every day. Every day. Seven days a week. When I used to come to the office before the pandemic, I did not have any walk routine. I am a bit fixated about my car, so I am reluctant to park it somewhere unsheltered to walk part of the way to work. Having a managerial position helped with that, because I have a reserved spot in the parking lot, so it was easy for me to have the habit of using the car instead of walking*”—Mike

**Table 6 ijerph-19-14598-t006:** Longitudinal analysis of reflexive thematic analysis.

Preretirement Theme/s	Postretirement Theme/s	Analysis
The discernment aspect of retirement.	Change is inevitable.	During the preretirement period, participants were considering the change which retirement was going to bring with it. There was a process of contemplation about what to do in the next phase of life, and the adaptation that needed to be carried out. They identified retirement as an opportunity to increase their PA behaviour. However, when people retired, they found the changes required to shift from a work routine to a retiree routine challenging. The adjustment required within the daily routine to fill all the time they used to spend at work with other activities was least thought-about in the preretirement period.
The discernment aspect of retirement + learned experiences.	Retirement is a choice.	The choice to retire is perceived to be an autonomous one that an individual makes based on personal circumstances. This choice seems to be based on experience and the results of the preretirement contemplative period. Based on perceptions of what retirement represented, participants consciously or unconsciously set their retirement-day target. However, this was part of the contemplation process in which they evaluated their life circumstances and the possible family and financial situation they would find themselves in. Through balancing these expectations with the actual life situation, participants took a conscious choice to retire. This conscious decision was not identified two years prior, even though participants were thinking of retiring. In effect, 35% of the participants decided to extend their working life by more than two years.
The inevitable process of ageing.	Retirement brings about reflections about ageing.	A theme that was importantly identified by the participants in the pre- and postretirement period was the ageing process. In the preretirement period, participants were becoming more aware of their ageing bodies, but they were still distracted by their work routine. The concept of—and perceptions on—ageing had started to kick in as a result of the retirement transition. However, once they retired, there was a heightened reflection on ageing, as they had more time for themselves. The participants felt that reflections about ageing after retirement made them change their outlook towards life.
Engagement in PA and learned experiences.	Retirement results in adjustments of physical activity.	An adjustment in the amount of PA that participants engaged in was similar to their preretirement perceptions. This perceived change seems to be dependent on their previous level of engagement with PA and their positive experiences when exercising. Based on their personal definition of PA, participants perceived a change in their PA engagement. This was also influenced by their previous PA and sitting time at their workplace.
	Retirement brings about lifestyle adjustments.	Lifestyle adjustments that participants claimed had happened once they retired had not been identified in the preretirement period. Some of the lifestyle adjustments were due to financial constraints, which had caused participants anxiety in the preretirement period. However, once they retired, they settled into a new financial routine without any mentioned concerns. The lifestyle adjustments identified by the participants have been carried out due to the increased free time identified when they retired. As a result of them having more free time, they adjusted their lifestyle accordingly. Changes identified were a change in smoking patterns, cooking, and food intake. Participants claimed to cook heathier meals and try alternative food, as they had more time. Others ate more due to easy food access during the day. These changes were not identified in their preretirement plan and were described as something unexpected by participants.
The discernment aspect of retirement + psychosocial factors shaping the retirement transition + learned experiences.	Retirement leads to the development of a new norm.	All participants—those who retired and those expecting to retire—were planning for their new norm. The adjustment process varied from their preretirement plan, which was based on experience and psychosocial factors. However, it was a common expectation that they needed to adjust to their new reality. The development of a new norm was present in the identified preretirement themes. It was an expected change of retirement.
Cognisant SB.	Sedentary behaviour is influenced as an impact of retirement.	The approach to SB seems to be impacted by going through retirement. In the preretirement period, participants were more aware and mindful of their sedentary activities. They tried to reach a balance between their PA and SB. This balance was not identified or present when participants retired. In the preretirement period, SB was associated with relaxation periods or attributed to the nature of work. Post-retirement, afternoons were associated with sitting activities or watching TV, as they had nothing better to do. The importance of being active was still present, as participants tried to be active during their day, but there seemed to be less effort to avoid SB.
Engagement in PA.	An unexpected pandemic influences the retirement transition.	Certain life situations, such as ill health of a family member or a global pandemic, are unexpected events within the retirement transition, which participants had to adjust to. The pandemic influenced social participation. In the preretirement period, participants mentioned the importance of peer support to engage in PA. Without that peer support during the COVID-19 pandemic, the amount of PA engagement could have been negatively influenced.

**Table 7 ijerph-19-14598-t007:** Participants’ stories using structural narrative analysis.

	Albert	Quotes	Lilly	Quotes
Reaching retirement age	Albert considers himself an active person. He is involved at his workplace and has a sense of belonging. At the same time, he is looking forward to retiring someday soon, due to difficulties at work. He has two routines, a workday routine, and a nonwork day routine, as he works on a shift basis. His work is mostly sedentary, and he finds it tiring due to his long hours. The lack of movement in his work is one of the things which makes it tiring, according to him. On his off day, he tries to be active by involving himself in different social activities to keep himself busy. He links his business with exercise, and considers himself very active. He used to train regularly when he was younger. This seems to give him the conviction that he can still be active. He still engages with different exercise activities, such as swimming and walking (although with less intensity than when he was younger). He attributes part of the decrease in activity to him being older. As he is getting older, he is thinking about his retirement and considering different retirement options.	(Talking about when he was younger) I was a cyclist myself. I did two seasons, three at my peak. But that’s 40 years ago, because I’m going to be 62. At my peak I used to do 50 miles today and 100 miles the next day”. (Now talking about the fear due to old age and him getting sick) “…but I’m afraid, I’m afraid, that if I get sick, and decrease my walking, and reduce certain activity”.	Lilly enjoyed her work, but she thought it impeded her from exercising. There was too much to do every day to fit in exercise as well. When she was younger, this was not such a problem because she was very energetic, but it has become harder as she aged. Now, if she has a lot to do in a day, she feels stressed. She attributes this to her ageing body. She feels her body is getting older, and her energy levels are lower and the recovery period after exertion is longer than before. Exercise has been part of her life since she was young. She has particular memories of exercising in her 20s and 30s, when she started exercising regularly. She used to be quick on her feet and it has always been in her to be active. She defines herself as someone who needs to be challenged and involved in various activities, so exercising forms part of that. Comparing herself to colleagues and other people she knows, she thinks she is more active and involved than them. She sees exercise as a way of keeping physically and mentally active. It is integral to her life.	“I hope not! I’ve been waiting and waiting mm so that I’ve always for the last 10 years if only I had more to do to come swimming more to, cycle, mhm I use to play Badminton by the way before as well. in fact, that is how I met. Because I left Malta at age 18 and I was introduced to playing badminton in and go back in 76 and we didn’t have any gym over here except at work and so I was one of those who was trying to get a work permit in those days. mhm to work in the UK and they said 1st question is do you play badminton? (laugh)”
Retirement plan	Albert wanted to retire to get rid of his work. It was too much for him to handle. Retirement was an opportunity. He has a plan about what to do when he retires to keep himself busy. He plans to go out, meet new friends, and continue to practise his hobbies.	(Talking about old age and PA) “then it’s 60, 70 it’s still not the same. Definitely not. but (.) much more than from now on I will be one of them myself for I am already there. (Talking about his retirement options) If they told me for example to join certain associations. For an example of a club. I have already been asked. They ask me we need someone like you (to do manual work). They told me to be able to organize certain things. That’s part of the activity (after retirement)”	Lilly was planning to retire within the next six months. She has been planning for retirement for the past ten years.	“I mean I can do more if I have the time. If I retired I am to do more. That’s my priority. Retirement is for me to do more. If my health allows me I will do more mhm r: do more? p: I want a balance in life. It’s not just activities I have to use my head. I go to. I’m constantly going into things that other people would not dream of!”
Retirement decision	The decision to retire has been taken.		With the COVID-19 pandemic looming near her retirement, Lilly decided to continue in employment, as they were asked to work from home. this provided her with the right opportunity to sort of start rehearsing for retirement, because she could get used to being at home for most of the day. She had the time to be more active and was able to better organise her day. She felt that her retirement started without her having to retire.	“yeah, so maybe it was more. Yeah, I mean I was itching to retire so that I have more time…too. Uh, for the outdoor. But I’m full of cold, but I will be going swimming today. Mhm I’m feeling better. If I’m feeling very tired and that I won’t go. But I’m feeling better still with a cold. But I would go swimming I believe. It will. (gesturing) It will help me clear it. I’ve done this before so (.)”
Retirement	On retiring, the initial days were like a holiday. He watched television and ate excessively, with minimal PA, until he realised that this will be his retired life, and so decided to get into a new routine. Even though Albert was happy with his retirement, it was not easy for him to fill his time. He was always active, but filling all the free time in retirement was difficult for him. He was planning on spending more time at his favourite social club and making new friends, but due to the COVID-19 situation, he had to cancel his plans. Meanwhile, a new opportunity came up when his family obtained a new pet dog, which became his best friend. This helped him become active again, and he started going to the fields often, spending time with the dog. Sporadically, he goes out for long walks with the dog. Since retiring, Albert has noticed that he is finding himself thinking more, and has become more worried about things, since he is less busy. Being busy during the morning is very important for him. He finds that having initiative is important, so he tries to find things to do and be active as much as possible. His role in the family has also changed. He has taken up more house chores because his wife is still working.	“The first day I was happy to leave, and I didn’t have anything to do, I didn’t know where to start, what I was going to do now. Watching television, yes now I can watch television. The first two three days I was watching television movies, I started feeling hungry and increased my food and I stopped reading and increased my food. I went to eat a plate of pasta at eight in the morning…. “(the realisation on the need to do something about retirement) “I can’t keep going like this, I’m going to become like a mule if I keep going like this, I gave up, I gave up and I started taking oats, only oats. As you don’t, have nowhere to go now you might have some kind of problem. Otherwise, next winter, we’ll see what’s going to happen, because it’s raining, I can’t go to the field because of the mud.” (Still unsure how it will work out in the future)	Even though retirement was something she looked forward to, it still brought fear with it. The change into something new was a challenge. Work provided a routine and motivation to be active. Being retired resulted in difficulty getting motivated.	“How was your first day of retirement and how did you feel? P: Very difficult to say. It is the first day to me. The first day was when. Mhm. The school was shut. Because of COVID. And that’s where my retirement started, really. I went back to school, working, only for. Uh, what a beginning of October. I think it was. It’s not. We went back up until Christmas time, so I only had. Well October November, December working three months working and I was relieving at a time but still the hours were in school but from January the year before, UM. To me that was my first day of retirement”.
				“Yeah, when did they pull? I was getting paid. But for me it’s just a matter of being paid the rest of it. It was as if I was. I was like a bird coming out. Get out of the cage. I just couldn’t have enough of the outdoor at this time”.
Adjustment	Albert feels he has settled in his new routine and is happy with the way things are. He does not wish to go to his previous plan of spending a lot of time at the social club. Albert feels he is performing enough PA, as his weight is stable.	(Highlighting the temporary adjustment) “At the moment I’m working on a project! What about this project? I have some paint for sure somewhere” (pointing to his hands). “Yesterday I painted with green and white, I’m making windmills now, if you want to order one (laughing). So far, I’ve done two, one normal to pass the time and another one which is more complex. But this one has more work it complicated”.	She developed her own way of keeping motivated and coping with change. Her day became less stressful, and she had more time to exercise. COVID-19 emptied the streets, so it was an excellent opportunity to exercise outdoors. However, now, health problems have started kicking in, which are limiting her ability to exercise. She needed surgery after developing hand problems. These make exercise harder. She admitted that she is spending more time sitting compared to before. Even though she is actively carrying out purposeful, mentally stimulating activities, these are all performed while sitting.	“It’s a new beginning. I enjoyed what I had, but now it’s time for a change. And it is not easy because you have to think. But I was preparing myself. I think when I spoke with you, I was already had a plan of things. And in fact, I didn’t do. A lot of it was fear that I was not going to be able to adjust. I’m going to have too, too much time in my hands and I’m going to be bored. And if Lilly is board, uh, and uh. So yes, uhm, it’s still there. Because I I got a phone call it is pending and that means I can go in there and look and it is a list of things. What I could be doing in case because then here (point to her head). It goes blank. It’s very hard to get motivated, and if it’s written down, at least you say OK, well then, I go. So that was on the worst scenario I would look at that and start”.

**Table 8 ijerph-19-14598-t008:** Table to show meta-inferences.

Type of Meta-Inference	Meta-Inference	QUAL	QUAN	Previous Literature
Agreement	a. Change in PA behaviour	The inevitable change is identified with retirement.	Statistical difference in domestic and work PA in retired vs. nonretired persons.	Changes in behavioural pattern [39].
	b. PA engagement is influenced by past PA	PA habits are part of the participants’ story.	Engagement in leisure time PA correlated with pre- and postretirement behaviour; differences presented themselves in retired and nonretired individuals; No difference in motivation pre- and postretirement.	Continuation of PA habits [14,16].
	c. Adjusting PA behaviour	New norm with retirement.	Difference in moderate PA behaviour.	There are changes in PA that vary with SES [10,40,41].
Dissonance *	d. Influence of retirement on leisure time PA	COVID pandemic could influence leisure PA engagement.	Leisure time PA engagement was different in the two groups but increased according to previous patterns.	Other factors not considered could have influenced this change together with COVID [41].
Diffraction ^	e. Changes in activity	New norm with retirement.	No difference in total PA pre- and postretirement.	Identified in other studies especially those using self-reported measures [11,12].
	f. Differences in sitting time	Retired participants claimed that they were sitting more.	Sitting time remained constant in the retirees but increased in those still employed.	There are variations in sitting time across genders and some of the changes start pre-retirement in males [42,43].

* Refers to data that were not consistent. ^ Refers to differences stemming from philosophical approaches.

## Data Availability

Additional data are available through corresponding authors.

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
