# Peer review of "Physical Activity and Sedentary Behaviour with Retirement in Maltese Civil Servants: A Dialectical Mixed-Method Study"

_ijerph, 2022, doi:10.3390/ijerph192114598_

Round 1

Reviewer 1 Report

Abstract

-        Please clarify the sequence of qual and quant

-        The conclusion is not based on the results. This is an observational study design, but the conclusion is about interventions. These are more the implications of the study – which can be described in the discussion – than the conclusion for your research question.

Intro

-        Retirement is introduced very succinctly. I suggest to provide more information about this, as well as a definition.

-        Line 36: this reference indicates that we currently do not know if these interventions are effective, so you may want to adjust your statement in this sentence accordingly, or include more appropriate refs for this statement.

-        Although the authors provide information about previous research, it would be helpful for the reader if they not only provide the information, but help the reader to synthesize this and see where the limitations/gaps are that are being addressed in the current study. For example, to introduce the topic of retirement and PA, the authors report 2 studies on perceptions, followed by a cohort study, followed by an SR. There is no concluding sentence to the paragraph tying this information together. As a reader, I am left with this rather fragmented information. Why are these studies reported? Are they representative for the work on this topic? How do they contribute to your rationale for your study? Do they point towards gaps, do they have limitations, etc.

-        I suggest to also introduce ‘predictors of change in PA/SB’ and write a para about what is known and what your study adds, as this is part of your RQ.

-        There are some typo’s, for example in lines 30 (influence), 46 (sedentary behaviour with retirement transition), 50 (greater).

Methods:

-        Line 86: it seems that the reference for the reliability of the Maltese version is missing.

-        Line 73, 75, 89, 129, 130, 133: for ease of reading, it would be very helpful to briefly explain the concept mentioned in these sentences to readers less familiar with them.

-        Please add how the Qual and Quant interact, as it now seems like two separate data collection procedures, whilst you indicate earlier it is a sequential design. Please clarify how one informs or builds on the other throughout the research process.

-        Please clarify the definition and operationalisation for retirement you used in your study.

-        Figure 1: I am concerned about the large drop out for the quant part and what this means for external validity, this is esp the case as you refer to ‘inferential statistics’ in line 123. You also refer to this in line 141.

-        Line 127: in doing so, you assign equal weight to the differences, regardless of baseline status. For example, a change of 30 minutes MVPA would be the same for 120-150 minutes as for 0-30 minutes. An option to avoid this would be to use for example regression/ancova with baseline values as covariate. In addition, the quant analyses are quite simple, not controlling for potential covariates. This may not have been possible because of the sample size, but it would be good to include this in the discussion as a limitation.

-        Line 138-139: this sentence is difficult to understand. Line 137-151: Because of the jargon used, this was also not easy to understand. As indicated before, a brief description of these key terms would be informative.

-        Please add more detail about your qualitative analysis. It needs to be described in such detail that it can be replicated. Add e.g. by who was it done, multiple researchers, what software was used to aid, were interview transcribed, etc.

Results:

-        Line 158: non-respondents at follow up?

-        You do describe differences between responders and non-responders, but a basic description of your sample is missing. Please describe the characteristics of your quant sample, e.g. gender, education, marital status.

-        Please check if you reported appropriate descriptives. For example, the variables in Table 1 are highly skewed and the mean and SD are not appropriate. In contrast in table 2, mean differences are typically normally distributed. But please check this for your data and either report mean (SD) ore median (interquartile range). Also add to the legend if you compared between or within groups. In my view, it would be more logical to compare between groups, rather than within comparisons.

-        Line 183, theme 1: this paragraph is very difficult to understand. From my perspective, your message did not come across.

-        Line 191: Would it be better to change the title to ‘The retirement plan is influenced by CIRCUMSTANCES DURING the transition.’? Maybe my confusion is partly fed by my definition of retirement transition, therefore, a clear operationalisation for your study would be very helpful.

-        Tables 3-5: Subthemes are clearly described and well supported by quotes.

-        Table 6: It is not entirely clear to me of this is about patterns, or if these are examples. What is the added value of this information in the scope of the paper?

-        Figure 2: This seems a very important Figure in the paper, making optimal use of the MM. There are some typo’s in the Figure.

-        Table 7: I appreciate the effort to bring the qual and quant together. I think the qual in your paper is very interesting and informative. Although I am sure the intention was very good, I do have some questions about the quant because of the drop out and used statistics. It would therefore be good to include in your discussion something about for example, selection bias, selective drop-out and lack of adjustments for relevant covariates [socioeconomic status is very important, as associated with the direction of changes] in your quant analysis. In short, are you convinced the findings in quant are real findings, brought about by retirement and are these data representative for the population?  My strong suggestion would be to either leave out the quantitative data, as your paper adds to current knowledge just based on your qual. Alternatively, I would back up the statements from your quant in Table 7 with literature. For example, there are several systematic reviews describing changes in PA domains across retirement transition.

Discussion:

-          Throughout the discussion, there is a good integration of the qual and quant. However, it would be informative to further support the discussion by relevant literature. In addition, in the first paragraph (line 237-253), it is not entirely clear whether your findings are in line with previous studies or not.

-          The structure of the discussion seems to be the following: changes in PA domains – decision to retire – change in LTPA – changes in Pa intensities – covid and other influencing factors- changes in total PA and sitting time. Is this correct? This does not immediately seem intuitive to me, so some guidance in understanding this structure may be helpful for the readers to navigate this.

-          Line 240: You describe changes in several PA domains and intensities. Please also specify whether these are increases or decreases. The direction of changes is also missing in for example line 268 and in some other paragraphs throughout the results and discussion section.

-          Line 261-262: It might be better to rephrase: ‘… can be influenced by various factors such as psychosocial factors, financial situation and health status.’

-          Line 263 typo ‘necessary’ should be ‘necessarily’

-          Line 277-278 The second part of this sentence (‘they did not correlate in the retired participants.’) might not be immediately clear for the reader.

-          Line 280-282: It would be clearer to write ‘the difference between the mean changes in the groups was not statistically significant’. Similar suggestion for the next sentence.

-          Lines 293-299: Good that the impact of Covid is acknowledged here. There are indeed several factors influencing changes in PA and SB during the retirement transition, of which some have not been mentioned in the paper. In line with the first comment to add more literature to the discussion, I suggest to check this and to also add other potential important factors that the QUANT analyses in your paper have not taken into account.

-          Line 303-304: ‘The different patterns being created might be related to the type of activity being carried out but not to activity behaviour.’ The difference between ‘type of activity being carried out’ and ‘activity behaviour’ is not immediately clear. A suggestion is to rephrase into ‘… but not to the amount of activity being carried out’.

-          Line 304-305: The statement ‘Exercise behaviours have little purpose for retired older adults’ is in my opinion too generalizing. The paper you refer to does not state that all exercise behaviours had little purpose for older adults.

-          Line 320: Tools were available in two languages to include participants from low socio-economic class. Did this work? Most participants had low PA jobs prior to retirement (line 289). Some more information about the sample should be included, as socio-economic status is an important factor here.

-          I suggest to add as a limitation that the analyses were not adjusted for other relevant factors associated with the outcome, such as gender, socioeconomic status.

-          In your practical application you write an advice for the Department of Health in Malta. I think this is less relevant for an international journal.

My suggestion is major revision, this is predominantly based on the amount of edits. In themselves the edits and requested changes are not major, except for the comment about potentially leaving out the quant part and the need to add more literature to the introduction and discussion. 

Reviewer 2 Report

Thank you for the opportunity to review your manuscript. You have delivered the current topic.

I have several recommendations for you:

1.      Revise your abstract according to the central theme and findings of your study

2.      Introduction: Your introduction is too short and ambiguous. Revise the introduction section and add subheadings according to your study objectives. The following three articles will help redesign the introduction section

1.      Saqib, Z. A., Dai, J., Menhas, R., S. M., Karim, M., Sang, X., & Weng, Y. (2020). Physical Activity is a Medicine for Non-Communicable Diseases: A Survey Study Regarding the Perception of Physical Activity Impact on Health Wellbeing. Risk Management and Healthcare Policy, 13, 2949. https://doi.org/10.2147/RMHP.S280339

2.      Menhas, R., Dai, J., Ashraf, M.A., M Noman, S., Khurshid, S., Mahmood, S., Weng, Y., Ahmad Laar R., Sang, X., Kamran, M., Shahzad, B., Iqbal, W., (2021). Physical Inactivity, Non-Communicable Diseases and National Fitness Plan of China for Physical Activity. Risk Manag Healthc Policy; 14:2319-2331. https://doi.org/10.2147/RMHP.S258660

3.      Yang, J., Menhas, R., Dai, J., Younas, T., Anwar , U., Iqbal, W., Ahmed Laar, R., Muddasar Saeed, M., (2022).Virtual Reality Fitness (VRF) for Behavior Management During the COVID-19 Pandemic: A Mediation Analysis Approach. Psychol Res Behav Manag ;15:171-182. https://doi.org/10.2147/PRBM.S350666

3. Add a subheading for the research statement, and under this, elaborate your study’s research questions

4. Materials and Methods

Revise your methodology clearly and comprehensively. Your current methodology is a mixture without logical reasoning.

i. Study used a mixed method based upon dialectical philosophy. What’s the dialectical philosophy in Mixed Method? How did you relate dialectical philosophy with the mixed method?

ii. The Exploratory Sequential MM design gave equal importance to the qualitative (QUAL) and quantitative (QUAN) strands QUAN. I suggest using QUAL and QUANT as a hybrid approach.

iii. Is there any pre-testing of the data collection tool? If yes, so provide the details

5. Revise your conclusion and makes it more scientific 

Round 2

Reviewer 1 Report

Thanks for the opportunity to review a revised version of this manuscript. We appreciate the authors’ effort in addressing our comments. The addition of Figure 1 is helpful and the description of the design and qualitative analysis is now clear. We also understand the authors’ rationale to present the data including the quantitative aspect. In our view, this manuscript now can be accepted for publication.

·        Please add the number of participants (n) to the title of your tables and double check titles (Table 3 seems to appear twice).  

·        Comment version 1 à I suggest to also introduce ‘predictors of change in PA/SB’ and write a para about what is known and what your study adds, as this is part of your RQ. Authors reply: This is now added under the heading of determinants of PA and SB. This heading was used as the literature is comprehensive on determinants but as identified in the introduction there aren’t studies which look at predictors of change within the retirement transition which is a novel aspect of our work. à We do not fully agree with this statement. There are certainly studies and systematic reviews, including papers the authors refer to that specifically address change – and predictors of change - in PA and SB during the retirement transition. We agree that your manuscript adds a very interesting mixed-method approach to this.

·        Line 457: ‘As found by Vansweevelt et al. [10] in their systematic review SES can have an impact on the level of PA and SB undertaken after retirement.’ Please note that this SR is about change during retirement, not on the level of PA/SB after retirement. It would be good to clarify this.

·        Table: there seems to be a typo in the last row à ‘various’ -> ‘variations’

·        Typo line 376: participants’ -> participants
